# Inhibition of transcriptional regulation of detoxification genes contributes to insecticide resistance management in *Spodoptera exigua*
Bo Hu [1,2], Yuping Deng[1,2], Tao Lu[1,2], Miaomiao Ren[3], Kuitun Liu[4], Cong Rao[4], Hailiang Guo[4] & Jianya Su [4] ✉

Synthetic insecticides have been widely used for the prevention and control of disease vectors and agricultural pests. However, frequent uses of insecticides have resulted in the development of insecticide resistance in these insect pests. The resistance adversely affects the efficacy of insecticides, and seriously reduces the lifespan of insecticides. Therefore, resistance management requires new strategies to suppress insecticide resistance. Here, we confirm that *CncC/Maf* are the key regulators of various detoxification genes involved in insecticide resistance in *Spodoptera exigua*. Then, we develop a cell screening platform to identify the natural compound inhibitors of *CncC/Maf* and determine that sofalcone can act as a *CncC/Maf* inhibitor in vitro and in vivo. Bioassay results showed that sofalcone significantly enhanced the toxicity (more than 3-fold) of chlorpyrifos and lambda-cyhalothrin against *S. exigua* larvae. Finally, we demonstrate that sofalcone can greatly improve the susceptibility of *S. exigua* larvae to insecticides by inhibiting the activity of the ROS/CncC-dependent detoxifying enzymes and downregulating the expression levels of detoxification genes. *CncC/Maf* inhibitors can be used as broad-spectrum synergists to overcome insecticide resistance in pest populations. Altogether, our results demonstrate that reduced expression of detoxification genes resulting from suppression of transcriptional regulation of these genes contributes to controlling insecticide resistance, which provides a very novel and high-efficiency green resistance management strategy.

Application of pesticides promotes agricultural development to meet global food demand. However, their overuse has caused severe resistance[1] of insects to almost all classes of commercial insecticides in these insect pests[2–4]. Insects develop resistance to insecticides through two major mechanisms: (i) target-site insensitivity resulting from the mutations in the insecticide target genes[5], and (ii) metabolic resistance, which results from the enhanced expression or activity of enzymes that metabolize to less toxic and/or more soluble compounds that could be excreted faster[6]. New insecticides are often needed to deal with target resistance; however, the creation of new

insecticides faces significant challenges in design, synthesis, testing, and analysis[7]. Therefore, insecticide resistance management requires the development of new strategies for preventing metabolic resistance. The complex metabolic enzymes mainly include cytochrome P450 mono-oxygenases (P450s), carboxylesterases (CarEs) and glutathione S-transferases (GSTs)[8]. Overexpression of these detoxification genes has been proven to be involved in insecticide resistance in insects. For example, upregulation of *CYP6BQ9* results in the majority of deltamethrin resistance in *Tribolium castaneum*[9]. Similarly, a CarE gene plays an important role in

[1]Jiangsu Key Laboratory of Sericultural and Animal Biotechnology, School of Biotechnology, Jiangsu University of Science and Technology, Zhenjiang, China. [2]Key Laboratory of Silkworm and Mulberry Genetic Improvement, Ministry of Agriculture and Rural Affairs, Sericultural Scientific Research Center, Chinese Academy of Agricultural Sciences, Zhenjiang, China. [3]College of Environment and Chemical Engineering, Jiangsu University of Science and Technology, Zhenjiang, China. [4]Key Laboratory of Integrated Management of Crop Diseases and Pests (Ministry of Education), College of Plant Protection, Nanjing Agricultural University, Nanjing, China. ✉e-mail: sjy@njau.edu.cn

the metabolism of chlorpyrifos and leads to resistance to this compound in *Nilaparvata lugens*[10]. Finally, *GSTe2* confers resistance to DDT and pyrethroid in mosquitoes[11].

In arthropods, constitutive overexpression of transcription factors contributes to increasing the expression of detoxification genes responsible for metabolic resistance[12,13]. Whether inhibition of transcriptional regulation of detoxification genes is used as a new strategy to suppress insecticide resistance remains unknown. Among these regulators, transcription factor *Nrf2* is the primary factor responsible for regulating the expression of detoxifying and antioxidant genes in mammals[14]. Under normal conditions, Keap1 acts as an E3 ligase adapter to retain Nrf2 in the cytoplasm[8]. The Nrf2-Keap1 interaction is disrupted under oxidative stress conditions and Nrf2 translocates to the nucleus, where it heterodimerizes with Maf. Subsequently, the Nrf2-Maf complex binds to the antioxidant response element (ARE) in the promoter region of genes coding for detoxifying and antioxidant enzymes, including P450s, GSTs, UGTs, NAD(P)H:quinone oxidoreductase-1 (*NQO1*) and heme oxygenase 1 (*HO-1*), which synergistically increase the efficiency of cellular defense systems[15]. *Nrf2* ortholog in invertebrates is known as cap 'n' collar C (*CncC*) that plays a critical role in regulating the expression of detoxification genes involved in insecticide resistance[16]. *CncC* and *Maf* enhance the expression of *CYP392A28*, *CYP391B1* and *CYP391A1*, responsible for fenpropathrin resistance in the spider mite *Tetranychus cinnabarinus*[17]. Similarly, these transcription factors also control the overexpression of P450 genes associated with deltamethrin resistance in *T. castaneum* and imidacloprid resistance in *Leptinotarsa decemlineata*[18,19]. Therefore, these transcription factors may serve as new target genes to screen for inhibitors of detoxification genes for pest control.

Constitutive activation of *Nrf2* contributes to resistance to chemotherapy drugs in mammalian cancer cells[20,21]. Inhibitors of *Nrf2* are often used to improve the susceptibility of cancer cells to drugs. For example, brusatol reduces the expression of *Nrf2* and downstream genes, and increases the susceptibility of lung cancer cells to cisplatin[22]. Similarly, DMC (2',4'-Dihydroxy-6'-methoxy-3',5'-dimethylchalcone) reduces the expression of GST genes by inhibiting the expression of *Nrf2* and preventing Nrf2 nuclear translocation, and reverses drug resistance in hepatocellular carcinoma cells[23]. Finally, Retinoic acid decreases the expression of *Nrf2*-

dependent genes by blocking the binding of Nrf2 to the ARE enhancer in breast cancer cells[24]. However, the screening of inhibitors of *CncC/Maf* has not be conducted, and whether the potential inhibitors can be used for insecticide resistance management remain poorly known in insect pests. Sofalcone, a synthetic analog of sophoradin, is a type of natural phenol derived from the traditional medicinal herb *Sophora subprostrata*, with potent anti-inflammatory activity in the human colonic epithelial cells[25].

*Spodoptera exigua* is a widespread and polyphagous lepidopteran pest that seriously damages many cultivated crops and causes considerable economic agricultural losses[26–28]. Currently, chemical insecticides are the major method for the control of this pest, and excessive and frequent insecticide applications have led this pest to develop resistance to many insecticides[29]. Therefore, finding synergists acting on novel target genes for resistance management becomes very important. Here, a high-throughput cell screening platform for identification of the inhibitors of *CncC/Maf* was established. We confirmed that sofalcone served as an inhibitor of *CncC/Maf* in vivo and in vitro. Furthermore, we determined that sofalcone significantly increased larval sensitivity to insecticides and the underlying molecular mechanisms.

## Results

### Identification of sofalcone as *CncC/Maf* inhibitor using a *CncC*-luciferase reporter gene assay in the Sf9 cells

To rapidly identify the effective natural product candidates to enhance the toxicity of insecticides to insect pests, we constructed the pGL3-CncC-Core promoter plasmid containing the *CncC/Maf* binding sequence (Fig. 1A). These constructs were transfected into Sf9 cells, chlorpyrifos, lambda-cyhalothrin and an equal volume of DMSO (control) were added after 16 h post-transfection. The lambda-cyhalothrin- and chlorpyrifos-induced luciferase activities of the pGL3-CncC-Core promoter were 3.1-fold and 4.7-fold greater than that of the control, respectively, whereas both two insecticides did not affect the promoter activity of the control pGL3-Core (Fig. 1A), suggesting that the pGL3-CncC-Core promoter vector were successfully constructed and can be utilized to perform large-scale screening of inhibitors for blocking the promoter activity of *CncC/Maf*. As shown in Supplementary Fig. 1, sofalcone did not change the luciferase activity of pGL3-CncC-Core promoter construct, whereas the other twelve natural

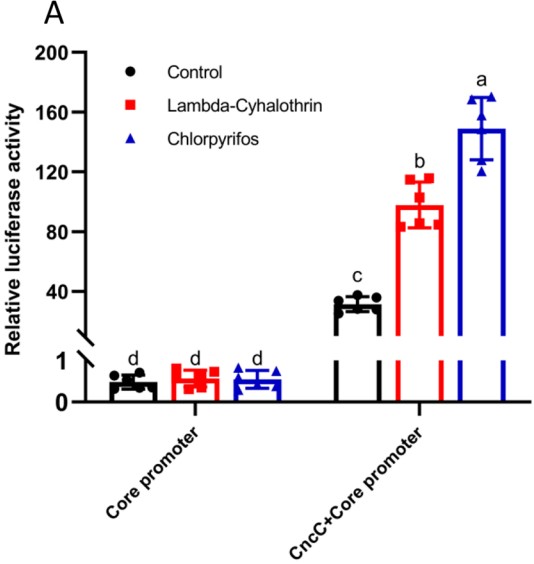

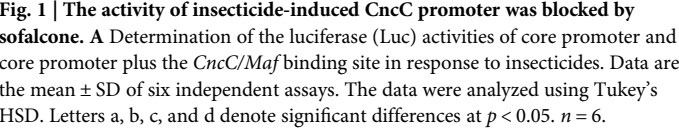

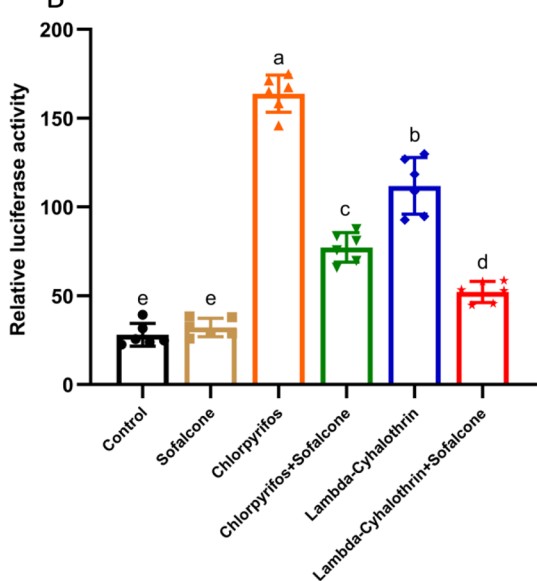

**Fig. 1 | The activity of insecticide-induced CncC promoter was blocked by sofalcone. A** Determination of the luciferase (Luc) activities of core promoter and core promoter plus the *CncC/Maf* binding site in response to insecticides. Data are the mean ± SD of six independent assays. The data were analyzed using Tukey's HSD. Letters a, b, c, and d denote significant differences at $p < 0.05$. $n = 6$.

**B** Promoter activities of PGL3-core-CncC/Maf after exposure to insecticides with or without sofalcone. Error bars represent the mean ± SD values ($n = 6$). Different letters on the right of error bars indicate significant differences based on ANOVA with Tukey's HSD multiple comparison test ($p < 0.05$).

products significantly enhanced the CncC-Core promoter activities compared with the control. Interestingly, pretreatment with sofalcone dramatically inhibited the insecticide-induced promoter activities of the CncC-Core sequence by 63.8% and 71.2%, respectively (Fig. 1B). Meanwhile, MTT assay results showed that sofalcone exposure (2.5 μM) did not affect cell viability (Supplementary Fig. 2). These results indicate that sofalcone do not affect the basal promoter activities of target genes of *CncC/Maf*, but significantly suppresses the insecticide-induced promoter activities of these genes.

### Sofalcone inhibited the expression of detoxification genes after exposure to insecticides in the *S. exigua* larvae

To investigate whether sofalcone affects the expression levels of detoxification genes after exposure to insecticides in vivo, the relative expression levels of these genes were analyzed in the *S. exigua* larvae exposed to DMSO, sofalcone, insecticide (chlorpyrifos or lambda-cyhalothrin) or in

combination. The results showed that the transcription levels of six CYP genes (*CYP321A8*, *CYP321A9*, *CYP321A16*, *CYP321B1*, *CYP9A11* and *CYP9A27*) were significantly overexpressed by 2.1- to 31.3-fold after exposure to each of the two insecticides, whereas the induced-expression levels of these CYP genes are significantly reduced by 31.6% - 100.0% in the larvae exposed to a mixture of insecticide and sofalcone (Fig. 2). Similarly, four GST genes (*GSTe2*, *GSTe6*, *GSTe14* and *GSTo2*) were significantly upregulated by 1.6- to 15.7-fold after exposure to each of the two insecticides, and pretreatment with sofalcone also inhibited the insecticide-induced expression levels of these GST genes by 40.7–100.0% (Fig. 3). Meantime, pretreatment with sofalcone also reduced the insecticide-induced expression levels of two CarE genes (*CarE1* and *CarE5*) and two ABC genes (*ABCB2* and *ABCC2*) by 43.8–100.0%, whereas the other two CarE genes (*CarE2* and *CarE4*) and *ABCB1* were not significantly affected in response to insecticides (Fig. 4 and Supplementary Fig. 3). *ABCC1* and *ABCC3* were clearly upregulated by 2.4- to 3.5-fold in the presence of

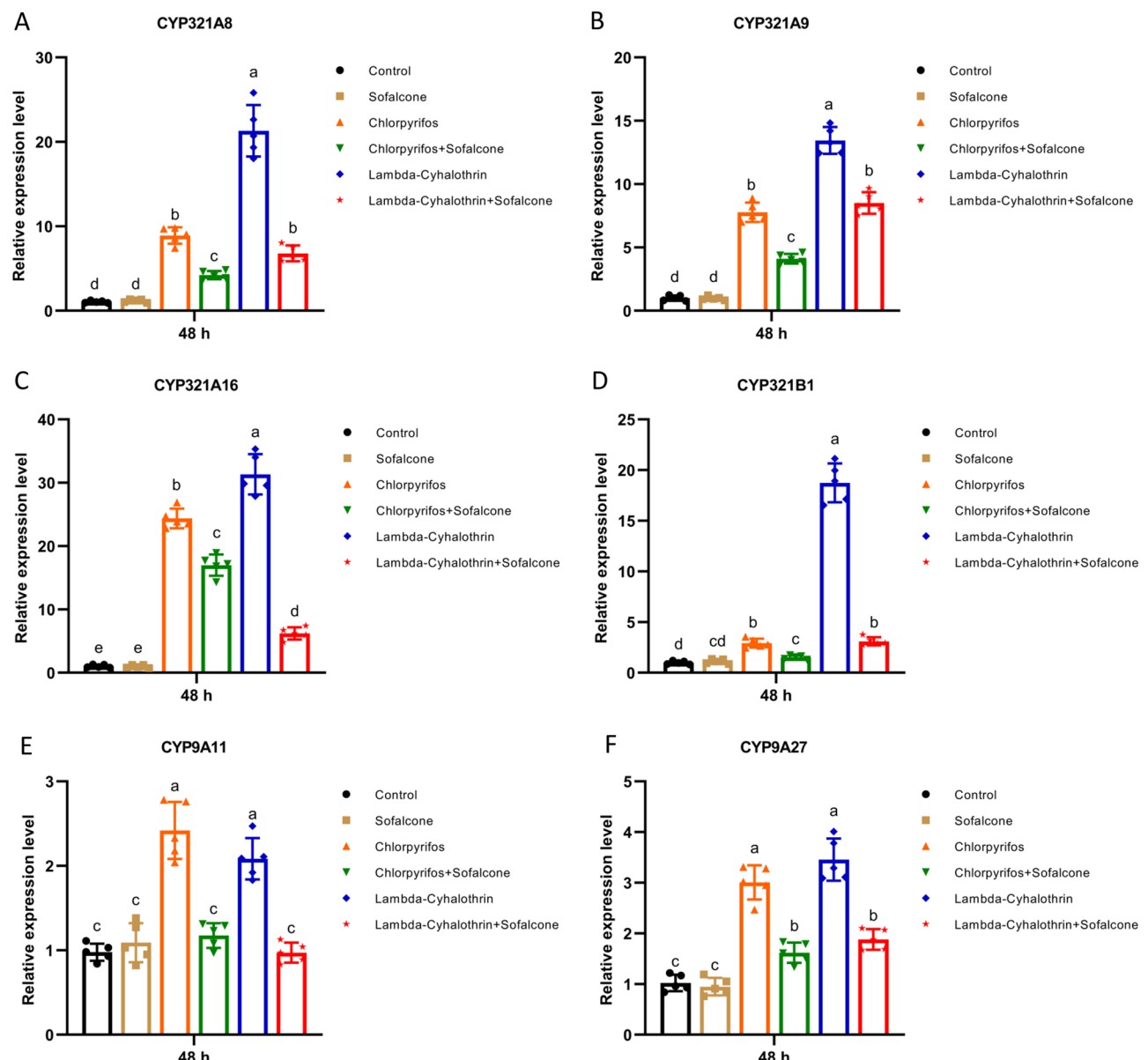

**Fig. 2 | Sofalcone inhibited the expression of P450 genes after exposure to insecticides.** The expression difference of *CYP321A8* (**A**), *CYP321A9* (**B**), *CYP321A16* (**C**), *CYP321B1* (**D**), *CYP9A11* (**E**) and *CYP9A27* (**F**) in response to insecticides with or without sofalcone was determined by RT-qPCR. Data are the mean ± SD of five independent assays. The data were analyzed using Tukey's HSD. Different letters on the right of error bars indicate significant differences ($p < 0.05$). $n = 5$.

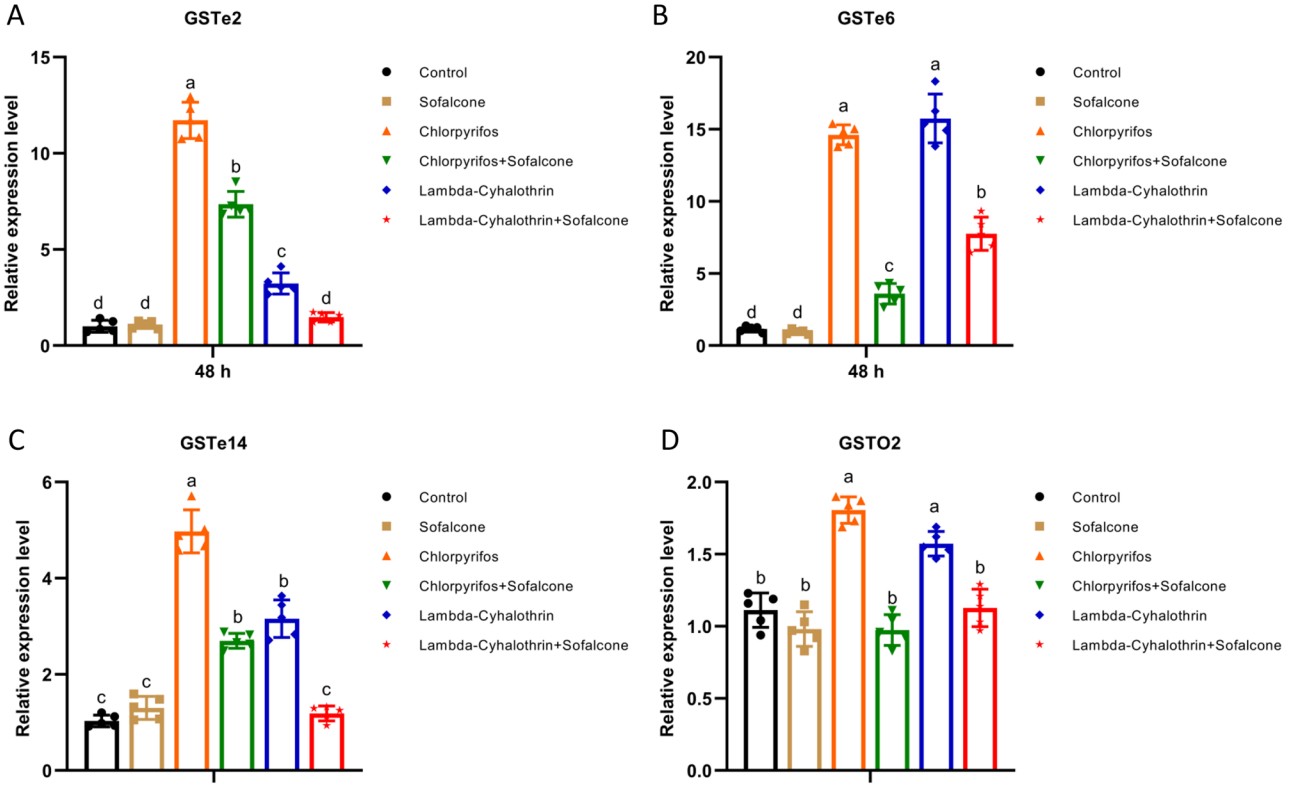

**Fig. 3 | The transcriptional expression of insecticide-induced GST genes was reduced by sofalcone.** RT-qPCR was performed to determine the difference in the expression of *GSTe2* (**A**), *GSTe6* (**B**), *GSTe14* (**C**) and *GSTO2* (**D**) in response to insecticides with or without sofalcone. Five biological replicates were conducted, and the $2^{-\Delta\Delta CT}$ method was utilized to calculate the relative expression levels. *GAPDH* and *β-Actin* were used to normalize expression levels. Letters on the standard error bars indicate significant differences compared with the control (ANOVA with post-hoc Tukey HSD, $p < 0.05$). $n = 5$.

insecticides, and pretreatment with sofalcone did not change the expression levels of the two genes compared to treatment with insecticides alone (Supplementary Fig. 3). These results suggest that insecticide-induced increases in the expression levels of detoxification-related genes were significantly downregulated by combined treatment with insecticides and sofalcone.

**The insecticide-induced activities of detoxification enzymes were repressed by sofalcone**

To determine the effect of sofalcone on detoxification enzyme activities, the assays of detoxification enzyme activities were performed in the larvae in the presence of insecticide with or without sofalcone. The results revealed that both chlorpyrifos and lambda-cyhalothrin increased the P450 enzyme activities by 2.0- and 1.7-fold, respectively, while cotreatment of insecticide with sofalcone dramatically reduced the insecticide-induced P450 activities by 62.5% and 59.1%, respectively (Fig. 5A). Similarly, treatment with chlorpyrifos and lambda-cyhalothrin resulted in a 2.0- and 2.2-fold increase in the GST activities, respectively, and the activities were significantly reduced by 60.3% and 72.2% after sofalcone and insecticide coappled treatment compared with treatment with insecticide alone (Fig. 5B). In addition, treatment with the sofalcone markedly decreased the lambda-cyhalothrin- or chlorpyrifos-induced esterase activities of *S. exigua* larvae by 69.6% and 52.1%, respectively (Fig. 5C). These results demonstrate that sofalcone effectively represses the increase in detoxification enzyme activities after exposure to insecticides in the *S. exigua* larvae.

**Sofalcone suppressed the expression of detoxification genes by *CncC/Maf* in the larvae exposed to insecticides**

To further explore the molecular mechanisms of the regulation of detoxification genes,

we cloned the upstream sequences of six CYPs, four GSTs, four CarEs and five ABCs by genomic walking. Transcription factor (TF) binding sites were predicted and analyzed using online tools JASPAR and ALGGEN (Fig. 6A, Supplementary Figs. 4–7 and Supplementary Table 5). According to the prediction of TF binding sites, four CYP450s (*CYP321B1, CYP321A16, CYP9A11* and *CYP9A27*), three GSTs (*GSTe2, GSTe6* and *GSTo2*), three CarEs (*CarE2, CarE4* and *CarE5*) and three ABCs (*ABCB1, ABCC1* and *ABCC2*) contained the AhR/Arnt binding site. The upstream sequences of three CYPs (*CYP321A8, CYP321A9* and *CYP9A27*), two GSTs (*GSTe2* and *GSTe6*), four CarEs (*CarE1, CarE2, CarE4* and *CarE5*) and four ABCs (*ABCB1, ABCB2, ABCC1* and *ABCC3*) possessed the binding sequence of ECR. PXR binding site existed in the promoter sequences of *CYP321A9, CYP321B1, GSTe6, GSTe14, CarE1, CarE5, ABCB1, ABCB2* and *ABCC3*. The binding sites of CncC/Maf are observed in the promoter sequences of detoxification-related genes including six CYP450s (*CYP321A8, CYP321A9, CYP321A16, CYP321B1, CYP9A11* and *CYP9A27*), four GSTs (*GSTe2, GSTe6, GSTe14* and *GSTo2*), two CarEs (*CarE1* and *CarE5*) and two ABCs (*ABCB2* and *ABCC2*). These results suggest that the transcription of these detoxification genes, which were significantly increased by insecticides, may be under the control of the same cis-regulatory element and transcription factors: *CncC* and *Maf*.

To determine whether the transcription factors *CncC* and *Maf* regulate the expression of detoxification genes, the fragment of *CYP321A9* putative promoter was ligated into the reporter gene vector and co-transfected with the *CncC* and *Maf* ligated into the expression plasmid. These results showed that the overexpression of Maf slightly enhanced the transcription activity of *CYP321A9* promoter, and CncC significantly increased the promoter activity of *CYP321A9* by 3.8-fold. Co-transfection with CncC and Maf resulted in a 6.5-fold increase in the promoter activity of *CYP321A9* (Fig. 6B). These results determine that the transcription factors *CncC* and *Maf* control the transcription levels of detoxification genes in *S. exigua*.

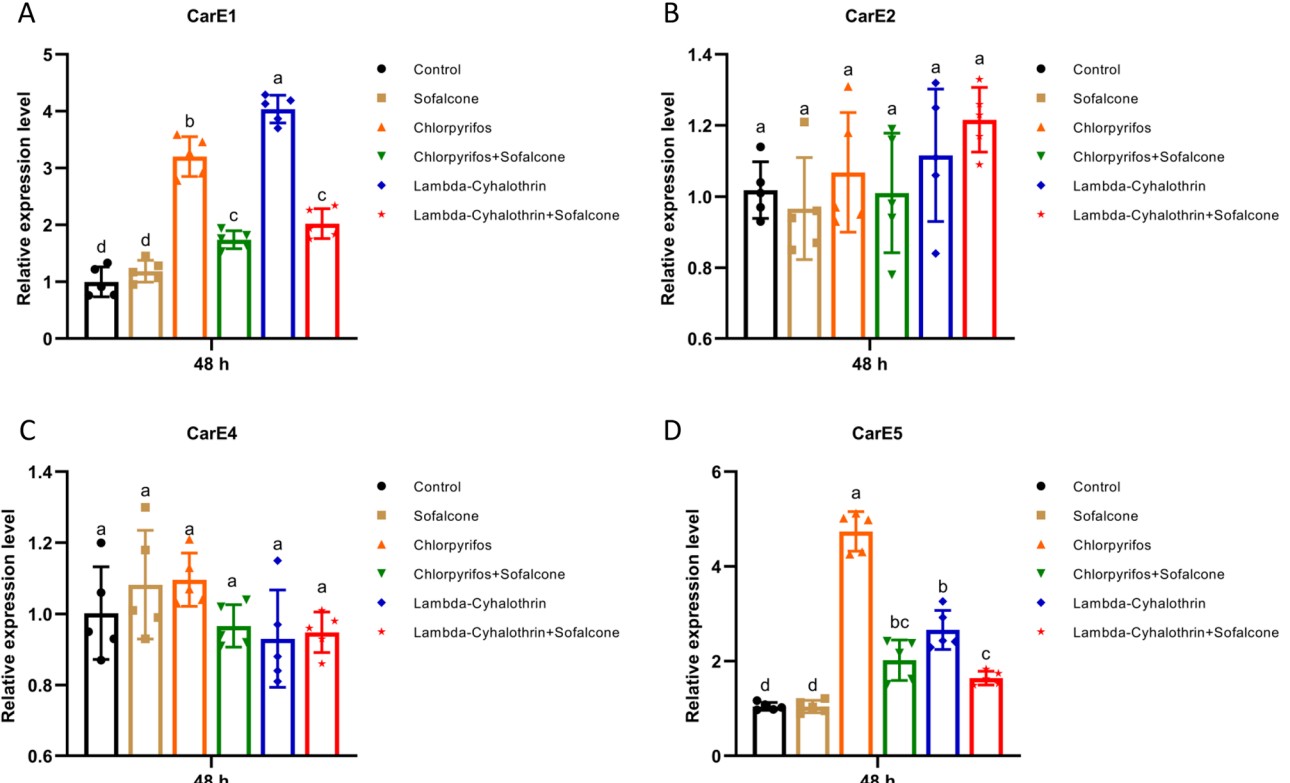

**Fig. 4 | Sofalcone diminished the mRNA levels of CarE genes when exposed to insecticides.** The expression changes of *CarE1* (**A**), *CarE2* (**B**), *CarE4* (**C**) and *CarE5* (**D**) under the stress of insecticides in larvae of *S. exigua* were determined by RT- qPCR. The results are presented as the mean ± SD ($n = 5$). Different letters on the error bar represent the significant difference based on ANOVA with Tukey's HSD multiple comparison test ($p < 0.05$).

To examine whether the expression of *CncC* and *Maf* was inhibited by sofalcone after exposure to insecticides in larvae, the mRNA levels of the two genes were analyzed by RT- qPCR. The results showed that the mRNA levels of *CncC* induced by chlorpyrifos or lambda-cyhalothrin were about 5.3- and 5.0-fold higher than that of control, respectively. Compared to the presence of insecticides alone, cotreatment with sofalcone and insecticides decreased the expression of *CncC* by 73.5% and 67.7%, respectively (Fig. 7A). Similarly, lambda-cyhalothrin- or chlorpyrifos-induced *Maf* expression was 3.1- and 2.9-fold, respectively. Also, sofalcone significantly prevented the lambda-cyhalothrin- or chlorpyrifos-induced upregulation of *Maf* (Fig. 7B). These results demonstrate that sofalcone concomitant treatment inhibits insecticide-induced expression of detoxification genes by transcription factors *CncC /Maf*.

## Sofalcone decreased the ROS content when exposed to insecticides

Our previous studies proved that the expression of detoxification genes was associated with oxidative stress induced by insecticides[30], so we measured the ROS content, including GSH-Px activity, hydrogen peroxide level and MDA content at 0, 6, 24, 48, and 72 h after exposure to insecticides with or without sofalcone. These results revealed that both chlorpyrifos and lambda-cyhalothrin significantly enhanced the GSH-Px activities at 6, 24, 48, and 72 h posttreatment, and the activity kept growing during the exposure period. Sofalcone prevented the increase in the GSH-Px activities in response to insecticides compared to independent treatment with insecticides (Fig. 8A, B). A significant increase in the production of hydrogen peroxide was observed in the larvae treated with chlorpyrifos for 48 or 72 h. A similar increase in the $H_2O_2$ levels was found after exposure to lambda-cyhalothrin. When challenged with sofalcone and insecticides, the insecticide-induced increase in the production of hydrogen peroxide was dramatically inhibited in the *S. exigua* larvae (Fig. 8C, D). Like the GSH-Px activity and $H_2O_2$ content, MDA also accumulated under chlorpyrifos stress

for 24, 48, or 72 h. Compared to the control, lambda-cyhalothrin treatment elevated the MDA content at 6, 24, 48 or 72 h. Similarly, co-treatments with sofalcone and lambda-cyhalothrin or chlorpyrifos dramatically decreased the MDA content (Fig. 8E, F). These results confirm that sofalcone blocked the insecticide-induced expression of detoxification genes through the ROS-dependent *CncC/Maf* in *S. exigua*.

## Synergistic effects of sofalcone with insecticides against *S. exigua*

To identify the synergistic effects of sofalcone on the *S. exigua* response to insecticides, sofalcone was coapplied with chlorpyrifos or lambda-cyhalothrin to third-instar larvae of the susceptible and resistant strains of *S. exigua*. The results showed that sofalcone significantly reduced the resistance of *S. exigua* to chlorpyrifos and lambda-cyhalothrin by 3.36- and 3.52-fold, respectively, whereas the mortalities of susceptible strain were not markedly increased by the cotreatment with sofalcone and insecticide compared to the treatment with insecticide alone (Table 1). Treatment with sofalcone alone did not affect the survival rate of larvae (Supplementary Fig. 8). These results demonstrate that sofalcone can be used as a synergist to increase the insecticidal effect of insecticides for agricultural and household use.

## Discussion

To cope with the stress of various toxic xenobiotics from the environment, insects have evolved a complex detoxification system, including cytochrome P450s, glutathione S-transferases and esterases. Increased expression of CYP genes has been proven to be implicated in resistance to insecticides in insects[31]. For example, the *CYP6BQ* genes are constitutively overexpressed in a deltamethrin-resistant strain of *T. castaneum* and result in resistance to this compound[18]. Similarly, transcriptional upregulation of four P450 genes *CYP9M10*, *CYP9J34*, *CYP9J40* and *CYP6AA7* leads to permethrin resistance in *Culex quinquefasciatus*[32,33]. In this study, all these CYP genes (*CYP321A8*,

**Fig. 5 | Sofalcone decreased the activities of detoxification enzymes when exposed to insecticides. A** P450 monooxygenase activity was evaluated by measuring ethoxycoumarin-O-deethylase (ECOD) activity. Significant differences ($p < 0.05$) in enzymatic activities are denoted using letters above bars (ANOVA with post-hoc Tukey's HSD). $n = 5$. **B** Glutathione-S-transferase (GST) activity was measured using 1-chloro-2, 4-dinitrobenzene (CDNB) as substrate. Different letters above the bars indicate significant differences based on ANOVA followed by post-hoc Tukey's HSD ($p < 0.05$). $n = 5$. **C** Esterase (EST) activity was determined by measuring a-naphthyl acetate activity. Different letters above the bars denote significant differences at $p < 0.05$ (ANOVA with post-hoc Tukey HSD). $n = 5$.

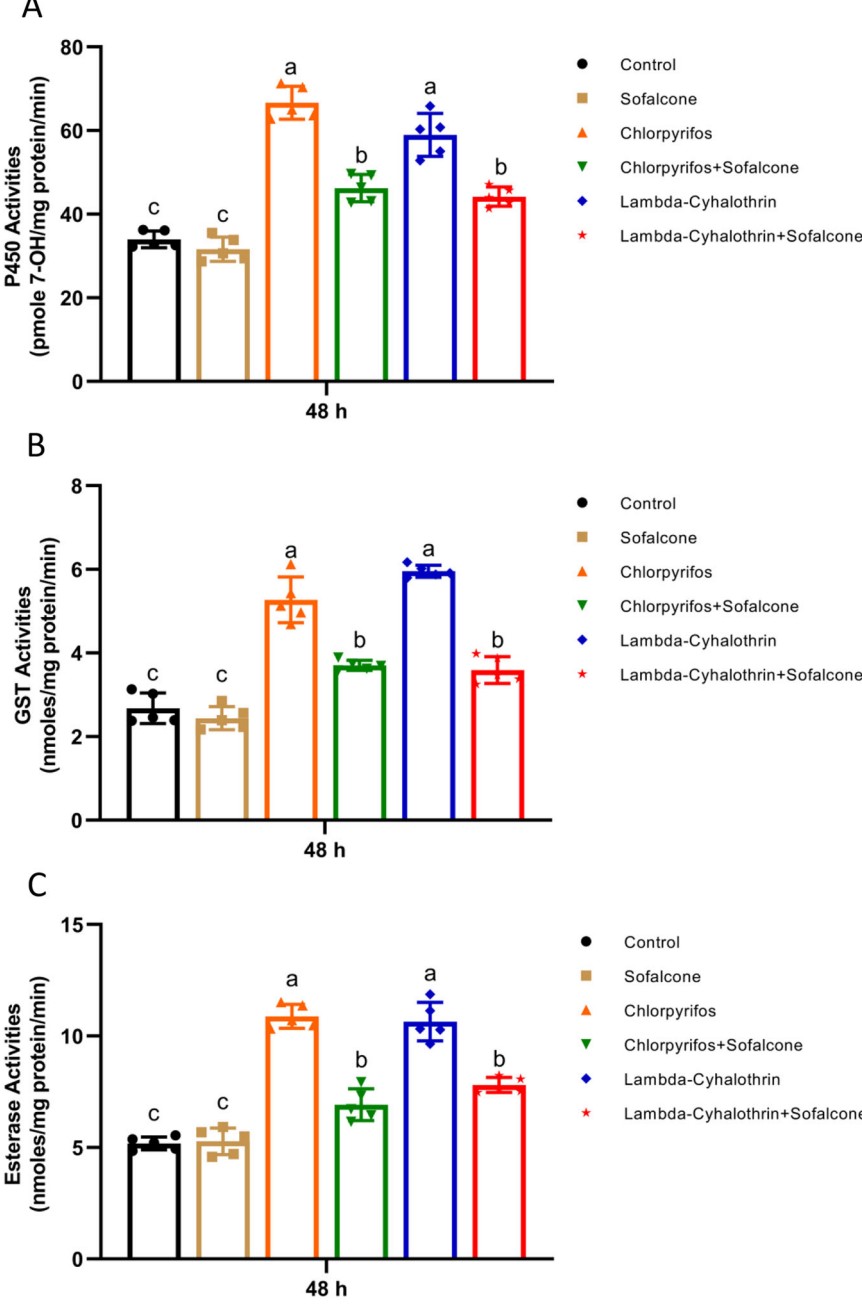

*CYP321A9*, *CYP321A16*, *CYP321B1*, *CYP9A11* and *CYP9A27*) were upregulated in the larvae of *S. exigua* upon exposure to lambda-cyhalothrin and chlorpyrifos. Our previous study has shown that the six CYP genes are constitutively overexpressed in the resistant strain used in this paper, and three of the six genes, *CYP321A8*, *CYP321A16* and *CYP332A1* have been proven to be associated with chlorpyrifos resistance[13,34]. Together with the results of our study, these findings illustrate that overexpression of multiple CYP genes is involved in resistance to insecticides in insects. Insect GSTs are a super-family of multifunctional enzymes that are involved in insecticide resistance[11,35,36]. In *N. lugens*, four GST genes, *GSTs1*, *GSTs2*, *GSTe1*, and *GSTm1*, are overexpressed in the imidacloprid-resistant strain, and the RNAi test indicates that the four GST genes contribute to the development of imidacloprid resistance[37]. Similarly, *GSTe2* and *GSTd1* play an important role in detoxification of thiamethoxam and fenpropathrin in Asian citrus psyllid *Diaphorina citri*[38]. Here we showed that *GSTe2*, *GSTe6*, *GSTe14* and *GSTo2* were significantly upregulated when exposed to insecticides. Our

previous work has shown that *GSTe6* and *GSTo2* are significantly upregulated in the resistant strain, and confer resistance to chlorpyrifos and cypermethrin[39]. These data suggest that enhanced expression of GST genes is associated with the development of insecticide resistance in insects. Carboxylesterases (CarEs) are important detoxification enzymes, which play a crucial role in the development of insecticide resistance in agricultural pests[40]. For example, upregulation of CarE genes results in indoxacarb resistance in *Spodoptera litura*[41] and overexpression of *CarE8* gene confers resistance to beta-cypermethrin and phoxim in *Plutella xylostella*[42]. In this paper, we revealed that both lambda-cyhalothrin and chlorpyrifos treatments significantly increased the expression levels of two CarE genes, *CarE1* and *CarE5*. These results suggested that the two CarE genes might play important roles in the detoxification of insecticides and the development of insecticide resistance in *S. exigua*. These studies, together with our findings, clearly demonstrate that upregulated expression of multiple detoxification genes confers resistance to insecticides in insects.

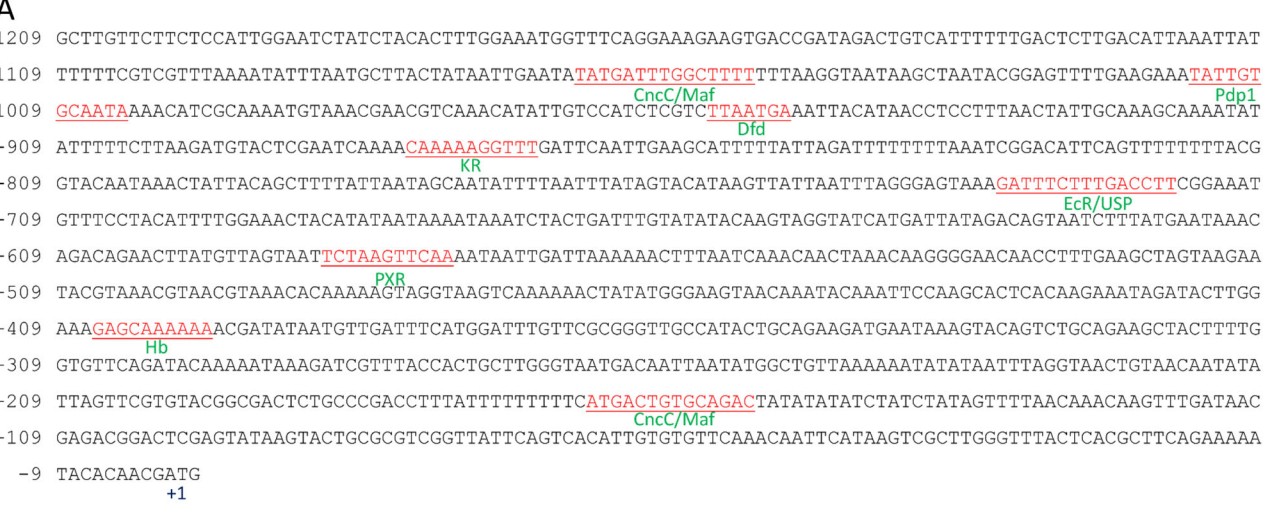

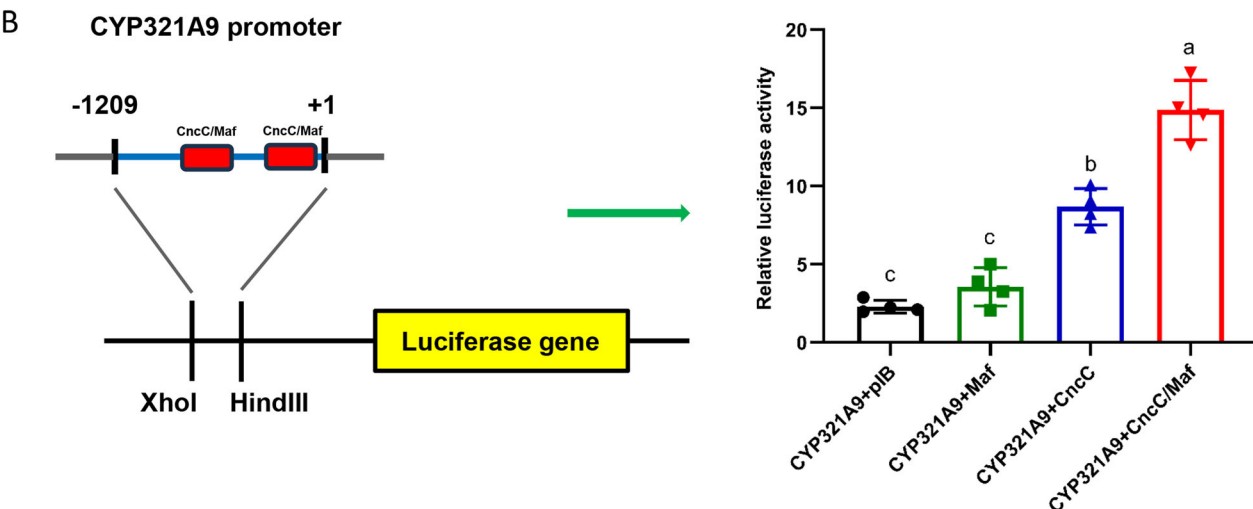

**Fig. 6 | *CncC/Maf* regulates the expression of *CYP321A9* gene. A** Prediction of transcription factor binding sites in a ~1.2 kb region of the promoter of *CYP321A9* gene. Nucleotides are numbered relative to the translation start site (ATG) indicated by +1. The predicted binding sites for transcription factors sites are underlined.

**B** The activity of each promoter construct in reporter gene assays in the presence or absence of CncC/Maf is shown. Error bars display SD. Different letters above the bars indicate significant differences based on ANOVA followed by post-hoc Tukey's HSD ($p < 0.05$). $n = 4$.

Although the importance of detoxification enzymes in conferring resistance to insecticides, the molecular mechanisms underlying the regulation of detoxification genes in insects are poorly understood[43]. In this study, our results showed that the binding site of *CncC/Maf* was present in the promoter region of detoxification genes, which were upregulated after exposure to insecticides. Furthermore, we determine that transcription factors *CncC/Maf* regulate the expression of *CYP321A9* gene. Our previous studies have proven that *CncC/Maf* are constitutively overexpressed in the resistant strain used in this study, and regulate the expression of multiple CYP and GST genes involved in insecticide resistance in *S. exigua*[13,30,34,39]. Here, the increase in the expression of *CncC* and *Maf* was also observed after exposure to insecticides. These results authenticate that transcription factors *CncC/Maf* mediate the expression of a series of detoxification genes associated with insecticide resistance in *S. exigua*. Similarly, *CncC/Maf* modulate the overexpression of *CYP6BQ* genes, which are responsible for deltamethrin resistance in *T. castaneum*[44], and increase the expression of *CYP392A28*, *CYP391B1* and *CYP391A1*, and result in fenpropathrin resistance in *T. cinnabarinus*[17]. Together these findings determine that *CncC/Maf* are the key regulators of various detoxification genes involved in insecticide resistance in insects. In mammals, the *Nrf2* is activated during the redox changes and regulates the expression of antioxidant and detoxifying

enzymes to maintain homeostasis[45]. Our previous work has shown that ROS activates the *CncC/Maf* pathway and enhances the expression of GST genes after exposure to insecticides[30]. In this study, both lambda-cyhalothrin and chlorpyrifos increased the ROS content, indicating that ROS mediates the expression of detoxification genes by the *CncC/Maf* pathway in *S.exigua*. Similarly, in *S. litura*, ROS elevates the transcriptional levels of *CncC* and *GSTe1* when challenged with chlorpyrifos[46]. These findings determine that ROS-activated *CncC/Maf* regulating the expression of detoxication genes in response to insecticides is conserved in insects[47,48]. Therefore, these transcription factors can serve as new target genes to screen inhibitors for pest control.

Insecticide application remains currently the major method for pest control. However, insect pests have developed serious resistance to multiple insecticides based on the Arthropod Pesticide Resistance Database (APRD 2016). Therefore, new ideas and methods are needed to delay insecticide resistance and extend the life of existing insecticides. In this study, a high-throughput cell screening platform for quickly identifying the inhibitors of *CncC/Maf* was established. Based on the platform, we found that sofalcone inhibited the insecticide-induced promoter activities of target genes of *CncC/Maf* in sf9 cells. Furthermore, our results revealed that sofalcone reduced insecticide resistance via inhibition of the expression of *CncC/Maf-*

**Fig. 7 | Sofalcone suppressed the expression of *CncC* and *Maf* after exposure to insecticides.** Relative expression of transcription factors *CncC* (**A**) and *Maf* (**B**) when exposed to insecticides in the presence or absence of sofalcone was determined by RT-qPCR. The $2^{-\Delta\Delta CT}$ method was used for the quantification of the gene expression level. The cycle threshold (Ct) values for the genes were normalized to the Ct values of *GAPDH* and *β-Actin*. Data are the mean ± SD of five independent assays. The data were analyzed using Tukey's HSD. Letters a, b and c denote significant differences at $p < 0.05$. $n = 5$.

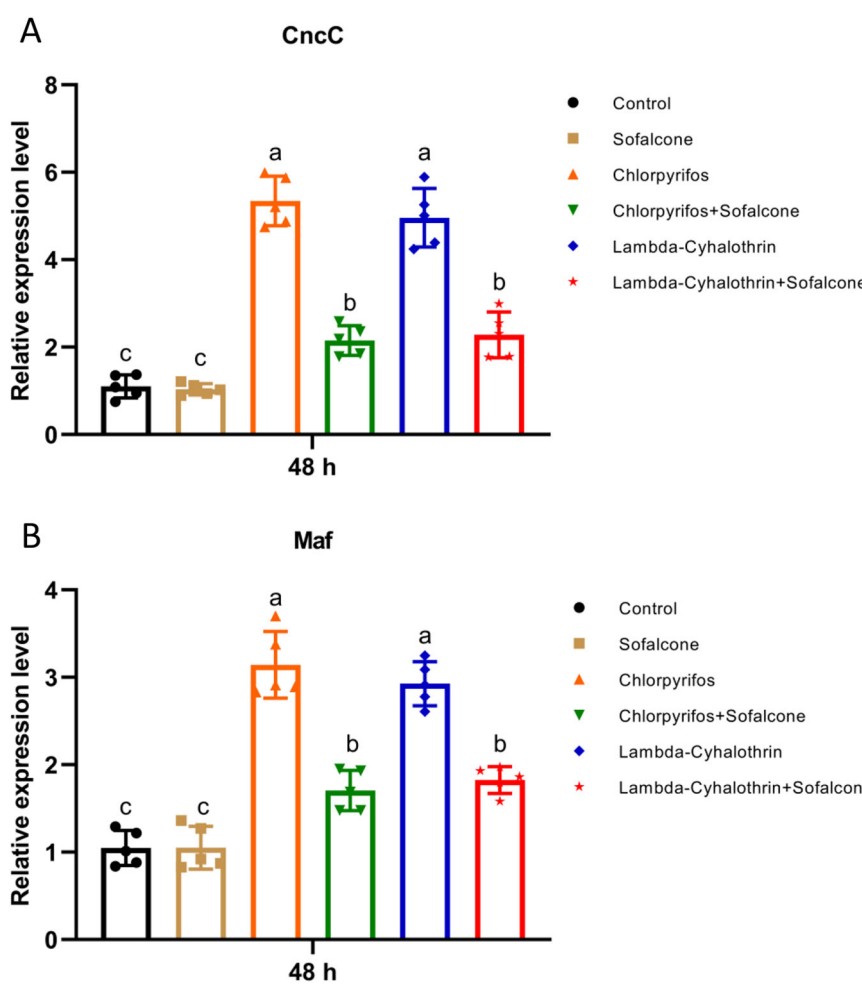

dependent detoxification genes in *S. exigua* larvae (Fig. 9). Sofalcone belongs to chalcone compounds, and its chemical structure is shown in Supplementary Fig. 9 according to these databases[49–51]. These results demonstrate that inhibitors of *CncC/Maf* can serve as synergists for insecticide resistance management. Although a cell line expressing ARE-luciferase is utilized to screen natural inhibitors of *Nrf2* pathway for controlling the chemoresistance of cancer cells[22], a similar approach has not been reported in insects. Such knowledge is important as it can help us establish cell screening platforms for the inhibitors and activators of the other transcription factors. Traditional synergists are often enzyme-specific inhibitors and their specificity limits their widespread application. For example, piperonyl butoxide (PBO), S,S,S-tributyl phosphorotrithioate (DEF) and diethyl maleate (DEM) are used as the enzyme inhibitors of P450s, CarEs and GSTs, respectively[2,52,53]. In this study, the insecticide-induced enzyme activities of P450s, CarEs and GSTs were significantly inhibited by sofalcone, suggesting that inhibitors of *CncC/Maf* have a broad-spectrum synergistic effect with insecticides in insect pests, which facilitates their widespread application. In this study, our results demonstrate that inhibition of transcriptional regulation of detoxification genes contributes to insecticide resistance management. Based on the innovative strategy, other inhibitors, and nucleic acid pesticides (dsRNA) can also be used to control insect pests. This method could delay insecticide resistance, extend the life of existing insecticides, and reduce pesticide usage and economic costs.

In conclusion, we confirm that *CncC/Maf* play a vital role in regulation of detoxification genes associated with insecticide resistance in a lepidopteran pest, *S. exigua*. A cell screening platform for identification of the inhibitors of *CncC/Maf* was developed. We determine that sofalcone can be used as a *CncC/Maf* inhibitor in vitro and in vivo. Furthermore, we present

evidence that sofalcone significantly inhibits the insecticide-induced expression levels and enzyme activities of ROS/CncC-mediated detoxification genes and further can greatly improve the susceptibility of *S. exigua* larvae to insecticides. Therefore, *CncC/Maf* inhibitors can function as broad-spectrum synergists in preventing insecticide resistance in insect pests. Our study provides an innovative and universal strategy to increase insecticidal activity and decrease application doses for sustainable crop protection based on the suppression of transcriptional regulation of detoxification genes.

## Methods
### Insects
The susceptible strain of *S. exigua* used in this study was obtained from Wuhan Kernel Bio-pesticide Company, Hubei, China. The resistant strain of *S. exigua* was collected in 2016 from Welsh Onion, *Allium fistulosum*, in Huizhou, Guangdong province, China.

According to our previous method[54], the resistant strain was selected for continuous generations by exposing neonate larvae to chlorpyrifos at a $LC_{70}$ concentration. Larvae were reared on an artificial diet at 25 °C under a 16-h light/8-h dark photoperiod with a relative humidity of 60 ± 5%[26,55].

### Chemicals
Lambda-cyhalothrin (95% TG) and chlorpyrifos (96.5% TG) were provided by Jiangsu Yangnong Chemical Co., Ltd, and Nanjing Red Sun Chemical Co., Ltd, respectively. Sofalcone, 4-hydroxychalcone and 2'-hydroxy-4'-methoxychalcone were purchased from Shanghai Merrill Co., Ltd. Lsoliquiritigenin, 2'-hydroxy-4,4',6'-trimethoxychalcone, trans-chalcone, cianidanol, 4-chlorochalcone, 4-nitrochalcone, chalcone and 4-fluorochalcone were purchased

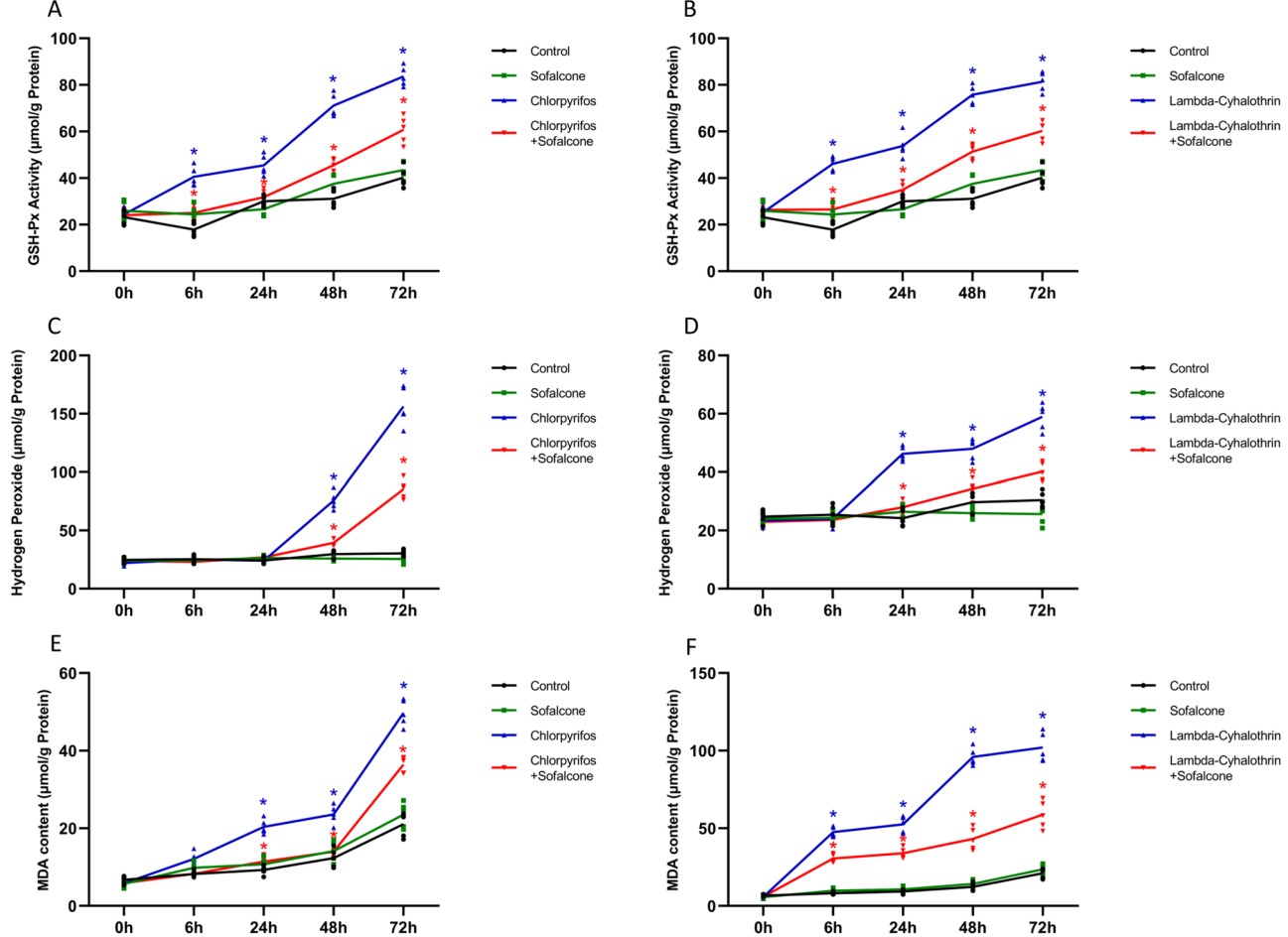

**Fig. 8 | Sofalcone decreased the ROS content when exposed to insecticides.** The activity of GSH-Px under the stress of chlorpyrifos (**A**) and lambda-cyhalothrin (**B**) with or without sofalcone was assessed by using a GSH-Px assay kit. $n = 5$. A Hydrogen peroxide assay kit was used to determine the activity of Hydrogen peroxide after exposure to chlorpyrifos (**C**) and lambda-cyhalothrin (**D**) with or without sofalcone. $n = 5$. The content of MDA exposed to chlorpyrifos (**E**) and lambda-cyhalothrin (**F**) with or without sofalcone was analyzed by using an MDA assay kit. The results are presented as the mean ± SD ($n = 5$). Blue asterisks on the standard error bars indicate significant differences between the treatment with insecticide only and control. Red asterisks represent significant differences between the combined treatment of insecticides and sofalcone and the treatment of insecticides alone. The significant difference was analyzed using a Student's $t$-test ($p < 0.05$).

**Table 1 | Synergism of sofalcone on the toxicity of insecticides against susceptible and resistant strains of *S. exigua***

| Strains | Insecticides | LC$_{50}$: mgAI/L (95% FL) | Slope ± SE | $\chi^2$ (df) | RR | SR |
|---|---|---|---|---|---|---|
| Sus | Chlorpyrifos | 1.880 (1.193 ~ 3.378) | 1.276 ± 0.316 | 0.448 (3) | | |
| | Chlorpyrifos + Sofalcone | 1.209 (0.746 ~ 2.234) | 1.079 ± 0.236 | 0.682 (3) | | 1.55 |
| Res | Chlorpyrifos | 1839.510 (1189.636 ~ 4129.388) | 1.333 ± 0.339 | 1.693 (3) | 978 | |
| | Chlorpyrifos + Sofalcone | 547.179 (358.932 ~ 809.619) | 1.365 ± 0.243 | 1.382 (3) | 291 | 3.36* |
| Sus | Lambda-Cyhalothrin | 1.515 (1.000 ~ 2.327) | 1.468 ± 0.315 | 0.559 (3) | | |
| | Lambda-Cyhalothrin + Sofalcone | 1.066 (0.604 ~ 1.854) | 1.120 ± 0.293 | 0.107 (3) | | 1.42 |
| Res | Lambda-Cyhalothrin | 2010.017 (1465.072 ~ 2827.967) | 1.855 ± 0.325 | 0.405 (3) | 1327 | |
| | Lambda-Cyhalothrin + Sofalcone | 571.217 (305.809 ~ 1210.014) | 0.764 ± 0.176 | 0.365 (3) | 377 | 3.52* |

*RR* resistance ratio, *SR* synergistic ratio, *Sus* susceptible strain, *Res* resistant strain.
*Significant difference with insecticide only without synergist.

from Sigma-Aldrich. Halofuginone and 4-methoxychalcone were purchased from Shanghai Aladdin Biochemical Technology Co., Ltd.

**Sample preparation**
Preliminary experiments were performed to identify the dose of synergists that showed no detrimental effects on 3rd instar larvae. One hundred milligrams per liter of sofalcone had no effects on larval survival and this

concentration was used in this study. The leaf dip method was used for sample preparation according to our previous studies[2,56]. The third-instar larvae of the resistant strain were transferred into plastic Petri dishes containing cabbage leaf discs supplemented with LC$_{30}$ concentration of insecticide (743 mg/L chlorpyrifos or 1048 mg/L lambda-cyhalothrin) with or without sofalcone and leaf discs that were treated with 0.1% Triton X-100 were used as the control. At least 20 third-instar larvae were exposed to each

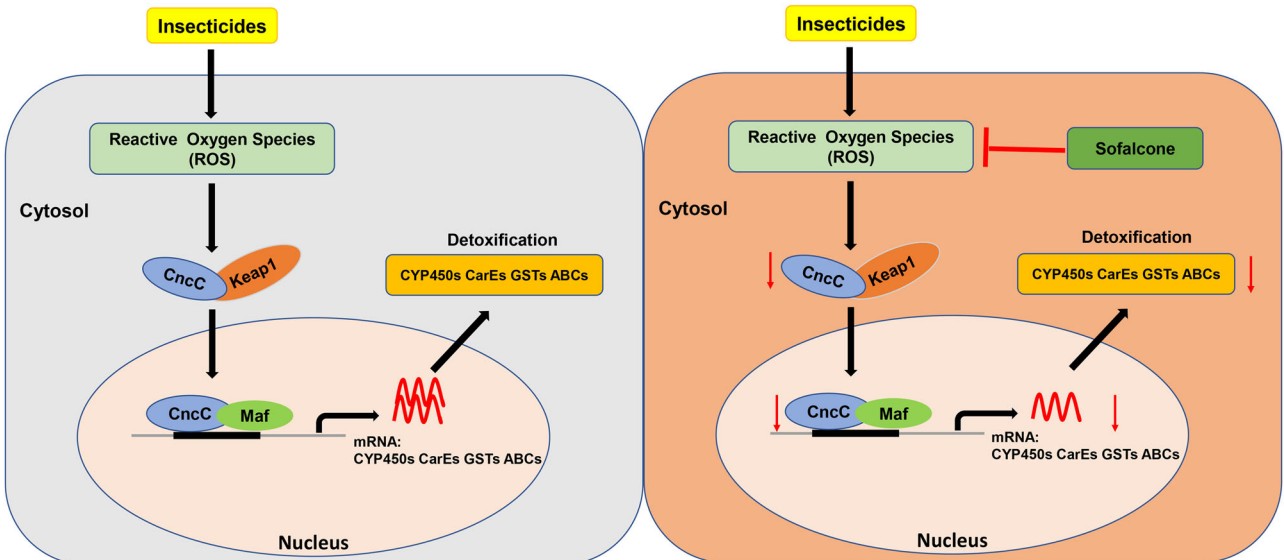

**Fig. 9 | Schematic of sofalcone as a synergist for insecticide management.** Insecticides induce ROS generation which activates the *CncC/Maf* pathway responsible for the expression of multiple detoxication genes associated with insecticide resistance in insects. Sofalcone can increase larval susceptibility to insecticides by inhibiting the activity of the ROS/CncC-dependent detoxifying enzyme and downregulating the expression levels of detoxification genes in insect pests.

treatment. Each replicate contained four larvae and five replicates were prepared for each treatment. After 48 h, the larvae were collected and stored at −80 °C for RNA isolation.

## RNA and DNA extraction

Total RNAs were isolated from the third-instar larvae of *S. exigua* using a TRIzol Reagent (Invitrogen, Carlsbad, CA, USA) following the manufacturer's protocol. The quality and quantity of total RNA were evaluated using a NanoDrop Spectrophotometer (Thermo Fisher Scientific, Wilmington, DE) and agarose gel electrophoresis, respectively. The cDNAs were synthesized using the HiScriptTM Q Select RT SuperMix Kit (Vazyme, Nanjing, China) according to the standard protocol. An Insect DNA Kit (Omega, USA) was utilized to extract the genomic DNA of larvae.

## Cloning of ABCs and CarEs

According to the transcriptome and genome data of *S. exigua*[27,57], putative sequences of one ABC and two esterase genes (*ABCB2*, *CarE1* and *CarE2*) were obtained, and the sequences of these genes (MK275645- MK275647) were verified by gene cloning and sequencing. All primers are shown in Supplementary Table 1.

## Quantitative real-time PCR (RT-qPCR)

RT-qPCR was performed using an ABI 7500 Fast Real-Time PCR System (Applied Biosystems). The primers were designed using Primer Premier 5 software (Supplementary Table 2). A 20 µL reaction mixture consisting of 10 µL of AceQ qPCR SYBR Green Master Mix kit (Vazyme, China), 1 µL of cDNA template, 0.8 µL of 10 µM forward and reverse primers was carried out. The PCR program was: 95 °C for 10 min, 40 cycles of 95 °C for 15 s and 60 °C for 40 s. Melting curves were added to determine the specificity of PCR products. *β-Actin* and *GAPDH* served as housekeeping genes[58]. All RT-qPCR analyses were performed in five biological replicates. The $2^{-\Delta\Delta Ct}$ method was utilized to calculate the relative expression levels of genes[59]. All methods and data were confirmed to follow the guidelines for the minimum information for publication of RT- qPCR experiments[60].

## Cloning and sequencing the 5'-flanking regions

Four restriction enzymes were used to digest the genomic DNAs for generating four pools according to the operation manual of the Universal Genome Walker Kit (Clontech, Palo Alto, CA, USA). The digested DNA fragments were ligated to Genome Walker adapters using T4 DNA ligase. Specific primers were designed to amplify the targeted sequences using LATaq polymerase (Takara, Japan). All primers are shown in Supplementary Table 3. The PCR products were ligated into the PMD-19 T vectors (Takara, Japan) and sequenced. The online software JASPAR (Relative profile score threshold > 80%) and ALLGEN (Maximum matrix dissimilarity rate >15%) were used to predict the transcription factor binding sites[61,62].

## Luciferase reporter assays

To screen the inhibitors of transcription factors *CncC/Maf*, the *CncC/Maf* binding sequence was subcloned into a CYP321A8-core-PGL3 vector containing the *CYP321A8* core promoter at the Xho I and Hind III restriction enzyme sites. The CYP321A8- (−142/-1) plasmid was used as the template[13]. To identify whether *CncC* and *Maf* regulate the expression of *CYP321A9*, the upstream sequence of *CYP321A9* gene (about 1200 bp) was ligated into the PGL3-Basic vector and the ORF of *CncC* and *Maf* was cloned into the pIB/V5-His vector. All primers are displayed in Supplementary Table 4.

Sf9 cells were cultured in sf-900 II SFM (Life Technologies, Carlsbad, USA) medium at 27 °C and an MTT assay Kit (Jiancheng, Nanjing, China) was used to evaluate the cytotoxicity according to the manufacturer's protocol. Briefly, the cells ($2\times10^5$ cells/mL) were seeded in 96-well culture plates and treated with sofalcone (1.25-10 µM) for 36 h. Then, the MTT was added into the cells and incubated at 27 °C for 4 h. Cell viability was measured according to manufacturer's instructions.

Sf9 cells were maintained in 24-well cell plates at a density of $4 \times 10^5$ cells. Then the 1 µg pGL3-CncC-Core promoter plasmid and 0.02 µg pRL-CMV were co-transfected to the cells using 2 µL FUGENE transfection reagent. After 16 h post-transfection, natural compounds (2.5 µM), chlorpyrifos (100 µM), Lambda-cyhalothrin (100 µM), or an equal volume of DMSO (control) was added. The cells were pretreated with 2.5 µM sofalcone for 4 h before the insecticide treatment. Similarly, 0.5 µg CYP321A9 promoter construct, 0.5 µg expression plasmid and 0.02 µg control pRL-CMV were co-transfected to the cells with the help of 2 µL FUGENE transfection reagent. Finally, the cells were collected after 48 h and luciferase activities were measured according to our previous method[39]. Each experiment was replicated four to six times.

**Article**

## Enzyme assays

Twenty 3rd-instar larvae were homogenized and microsomal fraction was obtained as described previously[63]. Each sample contained four larvae and five samples were prepared for each treatment. P450 monooxygenase activities were identified by measuring thoxycoumarin-O-deethylase (ECOD) activities on microsomal fractions as described previously by De Sousa et al.[64]. The microsomal fraction was resuspended and protein content was identified by the Bradford method[63]. The fluorescence was measured in a SpectraMax M5 multimode reader at 380 nm excitation, 460 nm emission and 30 °C for 15 min. P450 activities were expressed as mean picomoles of 7-OH per mg of microsomal protein/min ± SD.

GST enzyme activity was determined by measuring 1-chloro-2, 4-dinitrobenzene (CDNB) activity assay as described previously by Yang et al.[65]. 10 μL enzyme solution was mixed with 100 μL 1.2 mM CDNB (Sigma-Aldrich, USA) and 100 μL 6 mM glutathione (Sigma-Aldrich, USA) in each well. Enzyme activities were measured in a SpectraMax M5 multimode reader at 340 nm for 10 min (25 °C). Enzyme activities were expressed as micromole glutathione conjugate per min per mg of protein.

Esterase activity was measured using a-naphthyl acetate (a-NA) as a substrate according to the procedures developed by Yang et al.[65]. 10 μL enzyme solution and 200 μL substrate solution (0.1 mL 100 mM a-NA, 10 mg Coomassie Brilliant Blue G-250, and 5 mL 0.2 M pH 6.0 phosphate buffer) were added and mixed in each well. Enzyme activity was measured in a SpectraMax M5 multimode reader at 450 nm for 10 min (27 °C). EST activity is presented as nmole naphthol min$^{-1}$ mg protein$^{-1}$.

## Hydrogen Peroxide, MDA and GSH-Px assays

The hydrogen Peroxide assay was performed with a hydrogen Peroxide assay kit (Jiancheng, Nanjing, China). Twenty 3rd-instar larvae were homogenized on ice in a homogenization buffer, and centrifuged at $10,000 \times g$ for 10 min. Each sample contained four larvae and five samples were prepared for each treatment. One milliliter of supernatant solution was mixed with solution 1 and solution 2. The quantity of hydrogen peroxide was measured at 405 nm using a SpectraMax M5 multimode reader. The content of hydrogen Peroxide was presented as μmol/g protein.

An MDA assay kit (Jiancheng, Nanjing, China) was used to identify the MDA amount. The larvae were homogenized in PBS buffer on the ice. After the homogenization, the samples were centrifuged at 12,000 rpm at 4 °C for 15 min. According to the manufacturer's protocol, MDA content was detected at 532 nm using a SpectraMax M5 multimode reader. Results were expressed as μmol/g protein.

Similarly, the GSH-Px activity was quantified by measuring the decrease of glutathione at 412 nm using $H_2O_2$ as substrate following the manufacturer's guidebook (Jiancheng, Nanjing, China) by a microplate reader. The results of GSH-Px activity were expressed as μmol/g protein.

## Toxicological bioassay and synergism assay

The toxicity of insecticides to *S. exigua* was assayed by the leaf dip method as described previously[56]. Leaf discs cut from the cabbage were dipped into insecticide suspensions for 30 s. At least 30, 3rd-instar larvae were needed for each treatment and three independent assays were repeated. 100 mg/L of sofalcone had no effects on larval survival and this concentration was used in synergism assays. POLO-Plus Software 2.0 (Leora Software, Petaluma, USA) was used to estimate the $LC_{50}$ values and 95% fiducial limits (FLs). Synergism ratio was obtained by analyzing differences between $LC_{50}$ of insecticide alone and $LC_{50}$ of insecticide with the synergist[66].

## Statistical analysis

Statistical analysis was carried out using SPSS 16.0 software (SPSS Inc., Chicago, USA) and the data were presented as mean ± SD (standard deviation). Significant differences were calculated using a one-way analysis of variance (ANOVA) followed by Tukey's multiple comparisons test or Student's *t*-test. The level of significance was set at $p < 0.05$.

## Reporting summary

Further information on research design is available in the Nature Portfolio Reporting Summary linked to this article.

## Data availability

All data generated or analyzed during this study are included in the manuscript and Supplementary material. Supplementary material contains all primers used to amplify gene sequences and construct expression vectors. All the sequences have been deposited in GenBank (Accession nos: MK275645- MK275647). The source data behind the graphs in the paper can be found in Supplementary Data.

## Code availability

GraphPad Prism 8.0, PoloPlus and Excel were used to collect and analyze data.

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

## Acknowledgements

The work was supported by the National Natural Science Foundation of China (No. 32072452 and 32000333) and the Natural Science Foundation of Jiangsu Province (BK20221292).

## Author contributions

B.H. and J.S. designed the research. B.H., Y.D., T.L., M.R., K.L., C.R. and H.G. performed the experiments. B.H., Y.D., T.L. and M.R. analyzed the data. B.H. and J.S. wrote and revised the manuscript.

## Competing interests

The authors declare no competing interests.

## Inclusion & Ethics

Insects were used in this study and inclusion & ethics statement is not applicable to this study.
