## [Transparent Peer Review file · Communications Biology]

Inhibition of transcriptional regulation of detoxification genes contributes to insecticide resistance management in *Spodoptera exigua*

Corresponding Author: Dr Bo Hu

Version 0:

Reviewer comments:

Reviewer #1

(Remarks to the Author)

In this study, the authors identified a Cncc/Maf inhibitor sofalcone and explored the function of sofalcone in the controlling insecticide resistance of *Spodoptera exigua*. The methods used in this study were relatively reasonable, and the results displayed supported the conclusions. However, some comments for the authors to consider are listed as follows:

1. Line 371, the authors mentioned an artificial diet, the description needs to be supplemented in details.
2. The authors reported that 100 mg/L sofalcone had no effects on the larval survival of *S. exigua*. I wonder if other adverse effects of sofalcone could be observed, such as growth inhibition, etc.
3. In Figure 8, the statistical analyses were chaotic. What do the red and blue asterisks stand for? The comparison between the groups exposed to insecticide with and without sofalcone should be presented.
4. The authors defined sofalcone as the Cncc/Maf inhibitor, while in Fig. 9, sofalcone could be defined as the ROS inhibitor or scavenger. I think the authors should unify the definition of this compound. My personal preference is ROS scavenger.
5. So many formatting errors have been found and need further modification. For example, line 107 *Sophora subprostrata* should be *Sophora subprostrata*; line 230, including should be deleted; line 230-231, line 237, and line 241-243, 0h, 6h, 24h, 48h and 72h should be 0 h, 6 h, 24 h, 48 h, and 72 h; line 403, ABCB2, CarE1 and CarE2 should be ABCB2, CarE1 and CarE2; line 342, (DEM) should be (DEM); line 438, in in 96-well culture plates should be in 96-well culture plates; line 441, the should be deleted; line 476, g should be g; line 477, ml should be mL, line 503, Student's t-test should be Student's t-test; line 750, LC50 should be LC50; etc.

Reviewer #2

(Remarks to the Author)

the article is well written and presents very interesting results . i recommend its publication when minors corrections will be answered.

Please change everywhere CYP450 by CYP (for the gene and the RNA expression) OR by P450 (enzyme activities)

Line85: please explain what are NQO1 and HO-1

Please add in a figure the chemical structure of sofalcone and the reference of the compounds :

Human Metabolome Database HMDB0042013

PubChem Compound 5282219

PubChem Substance 175426956

ChemSpider 4445402

ChEMBL CHEMBL1441961

Line 111: please in addition to ref 27, use a more recent reference. Please be careful with the use of autocitation.

Line 133: in fig S1 could you explain the rational behind the use of different chalcone-like compounds?

Line 139: why are you using 2.5µM of sofalcone? Could you present the result of dose dependent experiment. In S2 only 4 dose where tested and I don't understand why you choose 2.5µM.

What is the link between LC50 in mgA/I and 2.5 µM. please use similar unit to facilitate the understanding of the reader

Line 197: please add a reference for ECR binding sequences as well as for PXR and all the binding sequence of fig6 (please add laso a table with the "binding site of all TF presented in this figure and in additional data, the table should include the position of the binding site as well as the statistical values associated with each binding site)

Line 264: please cite a review instead of 30-31-32.

Line 288: please cite a review instead of reference 41

Line 297-299: please cite the work of D amezian, G Le goff on Spodoptera frugiperda as they have published works of these transcription factor regulation of detoxification genes

Line 321-323: please cite a review at the end of this sentence

Line 352: please remove "protects the ecological environment as we don't know basic information of the sofalcone such as the half-life

Line 380: please what is Aladdin (add complete reference)

Line 383: please use molarity instead of mg/L

Line 387: please show the experiment linked to your choice of LC30 as a concentration for pesticide treatment in this article

line 388: why do you treat with 0.1% Triton X-100?

Line 406 and below: please explain how the cDNA were obtained?

File fig S4: please add the sequences of all binding motif identified in this figure with the pvalue or any other statistic associated with the discovery using JASPAR or ALLGEN. All the promoter cloned were obtained from susceptible or resistance *S exigua*?

please give the (and use the *S exigua* genome that is used as reference in database) reference of all the CYP, GST, esterase, ABC transporter in the *S exigua* genome. The name of these genes are often not correct in publication. Having the gene reference will avoid any mistake.

figS8: what is the concentration of sofalcone used? Could you explain what is the control?

Reviewer #3

(Remarks to the Author)

The emergence of insecticide resistance poses a significant challenge to the sustainable and efficient use of insecticides. The CncC/Maf-mediated expression of detoxification genes is a key mechanism driving metabolic resistance. In this study, Hu et al. developed a cell screening platform to identify sofalcone as inhibitors of CncC/Maf, and the inhibiting mechanism of sofalcone was investigated in *Spodoptera exigua*. Overall, the paper is well written, and the results presented in their work is clear. I have some minor points and hope that this feedback will assist the authors in improving their manuscript.

1. The P450, GST, and CarE gene families are large and commonly found in both vertebrates and invertebrates. Please introduce why these six P450s, four GSTs, and four CXE members were chosen for evaluation of gene expression in response to insecticides, with or without the presence of sofalcone.

2. Line 401-405: One ABC and two esterase genes (ABC2, CarE1 and CarE2) were verified by gene cloning and sequencing. Other detoxifying genes in this study were confirmed through gene cloning or in prior studies?

3. Fig 2-8. The susceptible insects were used? Please clarify this in the manuscript.

Other minor issues:

Line 107: *Sophora subprostrata* should be italic.

Line 212: "...of CYP321A9. (Fig. 6B)." => "...of CYP321A9 (Fig. 6B)."

Line 230: "...MDA content 0h" => "...MDA content at 0h"

Line 281: "...in Asian citrus psyllid" => "...in Asian citrus psyllid *Diaphorina citri*"

Line 369: Please detail how to rear the resistant strain of *S. exigua*, including the frequency of insecticide application.

Line 387: Please specify the concentration of LC30 for chlorpyrifos or lambda-cyhalothrin.

Line 806: "Letters above..." => "Different letters above..."

Version 1:

Reviewer comments:

Reviewer #1

(Remarks to the Author)

The authors have replied to the comments I raised in detail, and I recommend accepting this paper for publication.

Reviewer #2

(Remarks to the Author)

thanks for the corrections made to the manuscript.

Could you please correct these last 2 points:

Line 151 : Please replace CYP450 by CYP

Line 353-354: Please replace the sentence "This method delays insecticide resistance, extends the life of existing insecticides, and reduces pesticide usage and economic costs."

By "This method could delay insecticide resistance, extend the life of existing insecticides, and reduce pesticide usage and economic costs"

Reviewer #3

(Remarks to the Author)

The authors have addressed my issue.

Response to Reviewers:

Point by point list of our response (marked with #) to the reviewer comments:

Reviewers' comments:

Reviewer #1 (Remarks to the Author):

In this study, the authors identified a Cncc/Maf inhibitor sofalcone and explored the function of sofalcone in the controlling insecticide resistance of *Spodoptera exigua*. The methods used in this study were relatively reasonable, and the results displayed supported the conclusions. However, some comments for the authors to consider are listed as follows:

1. Line 371, the authors mentioned an artificial diet, the description needs to be supplemented in details.

#DONE. We have cited the relevant reference. L375-376.

Reference:

Jia B, Liu Y, Zhu YC, Liu X, Gao C, Shen J. Inheritance, fitness cost and mechanism of resistance to tebufenozide in *Spodoptera exigua* (Hübner) (Lepidoptera: Noctuidae). *Pest Manag Sci*. 2009 Sep;65(9):996-1002. doi: 10.1002/ps.1785. PMID: 19459181.

2. The authors reported that 100 mg/L sofalcone had no effects on the larval survival of *S. exigua*. I wonder if other adverse effects of sofalcone could be observed, such as growth inhibition, etc.

#DONE. No significant growth and feeding inhibition were observed.

3. In Figure 8, the statistical analyses were chaotic. What do the red and blue asterisks stand for? The comparison between the groups exposed to insecticide with and without sofalcone should be presented.

#DONE. We have added some descriptions about the representation of asterisks. L831-834.

4. The authors defined sofalcone as the Cncc/Maf inhibitor, while in Fig. 9, sofalcone could be defined as the ROS inhibitor or scavenger. I think the authors should unify the definition of this compound. My personal preference is ROS scavenger.

Because we also refer to relevant research on mammals, they defined similar compound as the Nrf2/ARE pathway inhibitor. So we also defined sofalcone as the Cncc/Maf inhibitor.

Reference:

Wei X, Mo X, An F, Ji X, Lu Y. 2',4'-Dihydroxy-6'-methoxy-3',5'-dimethylchalcone, a potent Nrf2/ARE pathway inhibitor, reverses drug resistance by decreasing glutathione synthesis and drug efflux in BEL-7402/5-FU cells. *Food Chem Toxicol*. 2018 Sep;119:252-259.

Tang YC, Chang HH, Chen HH, Yao JY, Chen YT, Chuang YJ, Chang JY, Kuo CC. A novel

NRF2/ARE inhibitor gossypol induces cytotoxicity and sensitizes chemotherapy responses in chemo-refractory cancer cells. *J Food Drug Anal.* 2021 Dec 15;29(4):638-652.

5. So many formatting errors have been found and need further modification. For example, line 107 Sophora subprostrata should be Sophora subprostrata; line 230, including should be deleted; line 230-231, line 237, and line 241-243, 0h, 6h, 24h, 48h and 72h should be 0 h, 6 h, 24 h, 48 h, and 72 h; line 403, ABCB2, CarE1 and CarE2 should be ABCB2, CarE1 and CarE2; line 342, (DEM)) should be (DEM); line 438, in in 96-well culture plates should be in 96-well culture plates; line 441, the should be deleted; line 476, g should be g; line 477, ml should be mL, line 503, Student' s t-test should be Student' s t-test; line 750, LC50 should be LC50; etc.

#DONE. We have revised these formatting errors. L107, L230-233, L237, L242-243, L408, L444, L446, L482, L483, L509, L756.

Reviewer #2 (Remarks to the Author):

The article is well written and presents very interesting results. I recommend its publication when minors corrections will be answered.

Please change everywhere CYP450 by CYP (for the gene and the RNA expression) OR by P450 (enzyme activities)

#DONE. Changed. L149, L188, L194, L264, L269, L272, L276, L305, L345, L347.

Line85: please explain what are NQO1 and HO-1

#DONE. We have added relevant descriptions. L85.

Please add in a figure the chemical structure of sofalcone and the reference of the compounds:

Human Metabolome Database HMDB0042013

PubChem Compound 5282219

PubChem Substance 175426956

ChemSpider 4445402

ChEMBL CHEMBL1441961

#DONE. We have added the chemical structure of sofalcone in Fig.S9. L335-337.

Line 111: please in addition to ref 27, use a more recent reference. Please be careful with the use of autocitation.

#DONE. We have added more recent references. L111.

Line 133: in fig S1 could you explain the rational behind the use of different chalcone-like

compounds?

#Many chalcones and chalcone derivatives can suppress the expression of NRF2 or inhibit its binding to ARE, thereby downregulating the expression of downstream genes. So we chose these compounds.

Reference:

Lim J., Lee S H, Su C., 2013. 4-methoxychalcone enhances cisplatin-induced oxidative stress and cytotoxicity by inhibiting the Nrf2/ARE-mediated defense mechanism in A549 lung cancer cells. *Molecules and Cells* 36(4):340-346

Wei X., Mo X., 2018. 2',4'-Dihydroxy-6'-methoxy-3',5'-dimethylchalcone, a potent Nrf2/ARE pathway inhibitor, reverses drug resistance by decreasing glutathione synthesis and drug efflux in BEL-7402/5-FU cells[J]. *Food & Chemical Toxicology*.S0278691518302035

Line 139: why are you using 2.5µM of sofalcone? Could you present the result of dose dependent experiment. In S2 only 4 dose where tested and I don't understand why you choose 2.5µM.

What is the link between LC50 in mgAI/l and 2.5 µM. please use similar unit to facilitate the understanding of the reader

#Under normal circumstances, the concentration unit for compounds used at the cellular level is µM. At the same time, bioassays with insecticides often choose mg/L as the concentration unit to facilitate better understanding for readers.

We selected 2.5 µM because this concentration does not affect the baseline activity of CncC promoter, but can block the insecticide-induced promoter activity without affecting the cell survival rate.

Line 197: please add a reference for ECR binding sequences as well as for PXR and all the binding sequence of fig6 (please add also a table with the "binding site of all TF presented in this figure and in additional data, the table should include the position of the binding site as well as the statistical values associated with each binding site)

#DONE. We have added a table with the "binding site of all TF presented in this figure and in additional data (Table S5). The criteria for prediction are also described in the Materials and Methods section. L430-431.

Line 264: please cite a review instead of 30-31-32.

#DONE. We have cited a review instead of 30-31-32. L265.

Line 288: please cite a review instead of reference 41

#DONE. We have cited a review instead of reference 41. L289.

Line 297-299: please cite the work of D amezian, G Le goff on Spodoptera frugiperda as they have published works of these transcription factor regulation of detoxification genes

#DONE. We have cited the relevant reference. L298-300.

Line 321-323: please cite a review at the end of this sentence

#DONE. We have cited a review at the end of this sentence. L322-324.

Line 352: please remove “protects the ecological environment as we don’ t know basic information of the sofalcone such as the half-life

#DONE. Removed. L353.

Line 380: please what is Aladdin (add complete reference)

#DONE. We have added relevant descriptions. L384-385.

Line 383: please use molarity instead of mg/L

Here, we mainly aim to clarify the concentration of sofalcone used in the article, so we recommend using mg/L. L388-389.

Line 387: please show the experiment linked to your choice of LC30 as a concentration for pesticide treatment in this article

According to previous research, the concentration of LC30 was selected for insecticide induction experiments, so we also chose this concentration.

Reference:

Nian X, Luo Y, Ye H, He X, Wu S, He Y, Wang D. Effects of Sublethal Concentrations of Insecticides on Survival and Reproduction of Two Bactrocera Species (Diptera: Tephritidae). J Econ Entomol. 2022 Oct 12;115(5):1539-1544.

Zeng X, He Y, Wu J, Tang Y, Gu J, Ding W, Zhang Y. Sublethal Effects of Cyantraniliprole and Imidacloprid on Feeding Behavior and Life Table Parameters of Myzus persicae (Hemiptera: Aphididae). J Econ Entomol. 2016 Aug;109(4):1595-602.

Nouri-Ganbalani G, Borzoui E, Abdolmaleki A, Abedi Z, George Kamita S. Individual and Combined Effects of Bacillus Thuringiensis and Azadirachtin on Plodia Interpunctella Hübner (Lepidoptera: Pyralidae). J Insect Sci. 2016 Sep 16;16(1):95.

line 388: why do you treat with 0.1% Triton X-100?

Because the pesticides used in this paper were the active ingredients, 0.1% Triton X-100 could act as a surfactant to increase the solubility of these pesticides in water.

Line 406 and below: please explain how the cDNA were obtained?

#DONE. We have added relevant descriptions. L403-404.

File fig S4: please add the sequences of all binding motif identified in this figure with the pvalue or any other statistic associated with the discovery using JASPAR or ALLGEN. All the promoter cloned were obtained from susceptible or resistance *S exigua*?

#DONE. We have added a table with the "binding site of all TF presented in this figure and in additional data (Table S5). The criteria for prediction are also described in the Materials and Methods section. L430-431. All the promoter cloned were obtained from resistant *S. exigua*?

please give the (and use the *s exigua* genome that is used as reference in database) reference of all the CYP, GST, esterase, ABC transporter in the *S exigua* genome. The name of these genes are often not correct in publication. Having the gene reference will avoid any mistake.

#DONE. We have cited the relevant reference. L407.

Reference:

Zhang B, Liu B, Huang C, Xing L, Li Z, Liu C, Zhou H, Zheng G, Li J, Han J, Yu Q, Yang C, Qian W, Wan F, Li C. A chromosome-level genome assembly of the beet armyworm *Spodoptera exigua*. *Genomics*. 2023 Mar;115(2):110571. doi: 10.1016/j.ygeno.2023.110571. Epub 2023 Feb 4. PMID: 36746219.

figS8: what is the concentration of sofalcone used? Could you explain what is the control?

#100 mg/L of sofalcone had no effects on larval survival and this concentration was used in this study.

Leaf discs that were treated with 0.1% Triton X-100 were used as the control.

Reviewer #3 (Remarks to the Author):

The emergence of insecticide resistance poses a significant challenge to the sustainable and efficient use of insecticides. The CncC/Maf-mediated expression of detoxification genes is a key mechanism driving metabolic resistance. In this study, Hu et al. developed a cell screening platform to identify sofalcone as inhibitors of CncC/Maf, and the inhibiting mechanism of sofalcone was investigated in *Spodoptera exigua*. Overall, the paper is well written, and the results presented in their work is clear. I have some minor points and hope that this feedback will assist the authors in improving their manuscript.

1. The P450, GST, and CarE gene families are large and commonly found in both vertebrates and invertebrates. Please introduce why these six P450s, four GSTs, and four CXE members were chosen for evaluation of gene expression in response to insecticides, with or without the presence of sofalcone.

In our previous study, we found that these six CYP450 and four GST genes were constitutively

overexpressed in the resistant strain. Four CXE genes were upregulated after exposure to insecticides in our previous experiments. Therefore, we selected these genes.

2. Line 401-405: One ABC and two esterase genes (ABCB2, CarE1 and CarE2) were verified by gene cloning and sequencing. Other detoxifying genes in this study were confirmed through gene cloning or in prior studies?

CYP450 and GST genes have been reported in our previous studies, while other ABC and esterase genes were downloaded from NCBI.

3. Fig 2-8. The susceptible insects were used? Please clarify this in the manuscript.

#DONE. The resistant insects were used. L390-391.

Other minor issues:

Line 107: *Sophora subprostrata* should be italic.

#DONE. We have changed it. L107.

Line 212: "...of CYP321A9. (Fig. 6B)." => "...of CYP321A9 (Fig. 6B)."

#DONE. Deleted. L212.

Line 230: "...MDA content 0h" => "...MDA content at 0h"

#DONE. Added. L230.

Line 281: "...in Asian citrus psyllid" => "...in Asian citrus psyllid *Diaphorina citri*"

#DONE. Added. L282.

Line 369: Please detail how to rear the resistant strain of *S. exigua*, including the frequency of insecticide application.

#DONE. We have added relevant descriptions and cited our previously-published papers. L373-374.

Line 387: Please specify the concentration of LC30 for chlorpyrifos or lambda-cyhalothrin.

#DONE. We have added the concentration of LC30 for chlorpyrifos or lambda-cyhalothrin. L392-393.

Line 806: "Letters above..." => "Different letters above..."

#DONE. Added. L807-808.

Response to Reviewers:

Point by point list of our response (marked with #) to the reviewer comments:

Reviewers' comments:

Reviewer #2 (Remarks to the Author):

thanks for the corrections made to the manuscript.

Could you please correct these last 2 points:

Line 151: Please replace CYP450 by CYP

#DONE. We have changed it. L151.

Line 353-354: Please replace the sentence “This method delays insecticide resistance, extends the life of existing insecticides, and reduces pesticide usage and economic costs.”
By “This method could delay insecticide resistance, extend the life of existing insecticides, and reduce pesticide usage and economic costs”

#DONE. We have changed it. L353-354.